# Regulation of Normal and Neoplastic Proliferation and Metabolism by the Extended Myc Network

**DOI:** 10.3390/cells11243974

**Published:** 2022-12-08

**Authors:** Edward V. Prochownik

**Affiliations:** 1Division of Hematology/Oncology, UPMC Children’s Hospital of Pittsburgh, Pittsburgh, PA 15224, USA; procev@chp.edu; 2The Department of Microbiology and Molecular Genetics, The University of Pittsburgh Medical Center, Pittsburgh, PA 15213, USA; 3The UPMC Hillman Comprehensive Cancer Center, Pittsburgh, PA 15232, USA; 4Pittsburgh Liver Research Center, University of Pittsburgh, Pittsburgh, PA 15224, USA

**Keywords:** ChREBP, hepatoblastoma, hepatocellular carcinoma, Mnt, Mxd, MondoA, Mlx, TXNIP

## Abstract

The Myc Network, comprising a small assemblage of bHLH-ZIP transcription factors, regulates many hundreds to thousands of genes involved in proliferation, energy metabolism, translation and other activities. A structurally and functionally related set of factors known as the Mlx Network also supervises some of these same functions via the regulation of a more limited but overlapping transcriptional repertoire. Target gene co-regulation by these two Networks is the result of their sharing of three members that suppress target gene expression as well as by the ability of both Network’s members to cross-bind one another’s consensus DNA sites. The two Networks also differ in that the Mlx Network’s control over transcription is positively regulated by several glycolytic pathway intermediates and other metabolites. These distinctive properties, functions and tissue expression patterns potentially allow for sensitive control of gene regulation in ways that are differentially responsive to environmental and metabolic cues while allowing for them to be both rapid and of limited duration. This review explores how such control might occur. It further discusses how the actual functional dependencies of the Myc and Mlx Networks rely upon cellular context and how they may differ between normal and neoplastic cells. Finally, consideration is given to how future studies may permit a more refined understanding of the functional interrelationships between the two Networks.

## 1. Introduction

a. Myc dysregulation in cancer: a brief history. Since its identification over 40 years ago as the human cellular ortholog of the transforming avian *v-myc* retroviral oncogene [1], the *MYC* gene’s aberrant expression has been documented in many human cancer types [2,3,4]. This dysregulation may occur as a result of recurrent chromosomal translocations, gene amplification, aberrant growth factor signaling or point mutations that stabilize Myc mRNA or protein and that tend to be affiliated with specific tumor types [2,3,4]. For example, *MYC* gene translocations are confined almost exclusively to certain subtypes of lymphoid neoplasms (largely, although not exclusively, of B cell origin) whereas gene amplifications are particularly prevalent in breast and ovarian cancers [4,5,6,7,8,9,10]. In contrast to this structure-based over-expression, *MYC* dysregulation in response to constitutive growth factor signaling is commonly encountered in many tumor types with Ras mutations and in most sporadic colo-rectal cancers owing to acquired mutations of the *APC* tumor suppressor that unleash constitutive β-catenin signaling [4,11,12]. Indeed, the ectopic, over-expression of *MYC* alone is sufficient to initiate and necessary to maintain tumor growth in numerous mouse models, including those of lymphoma, breast cancer, osteosarcoma and hepatocellular carcinoma [13,14,15,16].

b. The Myc Network: overview. Within a decade of *MYC*’s discovery, and coinciding with a period during which its central role in naturally occurring oncogenesis became established, the Myc protein was identified as a bHLH-ZIP transcription factor that was dependent upon and subject to regulation by a small collection of other bHLH-ZIP proteins (Figure 1) [2,17,18,19,20,21,22]. Guided by strict rules that dictate which interactions among its members are permissible, the “Myc Network” closely regulates the expression of its target genes in simple yet elegant ways, while simultaneously allowing for the fine tuning needed to modulate proliferative signals in response to changes in intra- and extracellular environments [9,20,22].

c. The Mlx Network: overview. A second family of bHLH-ZIP transcription factors has been described whose members bear structural and functional similarities to those of the Myc Network (Figure 1). 

This equally small “Mlx Network” was originally proposed to regulate a more focused collection of target genes that significantly overlaps the larger array of Myc Network targets [9,20,28,30,31,32,33,34,35,36,37]. The binding site similarities between Myc Network and Mlx Network members, their co-binding to one another’s sites and the sharing of some Network members allows for some genes to be co-regulated in ways that impart an even more nuanced degree of combinatorial transcriptional control depending on the identity of the cell and its proliferative and metabolic states (Figure 1) [9,38,39].

d. Cross-talk between the Myc and Mlx Networks balances energy production and consumption. The following sections discuss how the Myc and Mlx Networks, both individually and cooperatively, regulate a variety of important processes such as proliferation and translation in several cell types, most notably fibroblasts and hepatocytes. Particular emphasis is placed on how these two anabolic processes, which are among the cell’s most energy-demanding [40] are matched with oxidative phosphorylation (Oxphos) and glycolysis so as to achieve the proper balance between ATP consumption and production. Additionally discussed are recent insights into how cross-talk between the Myc and Mlx Networks and their sharing of common binding sites in target genes may serve important regulatory roles by allowing for gene expression to be coordinated with the availability of certain nutrients and other metabolites.

## 2. The Myc Network and Its Control of Gene Expression

a. Myc Network components. The Myc Network has long been perceived as regulating ~10–15% of genes [2,9,19]. However, under “pathological” conditions, i.e., when Myc levels rival those encountered in some tumors or experimental states of over-expression, it may impact many more genes if not the entire transcriptome as a result of binding to low-affinity sites [2,9,19,41,42]. Target gene activation by Myc is dependent upon its prior association with its obligate bHLH-ZIP partner Max, which facilitates DNA binding to consensus “E box” sites (CACGTG) and the histone acetylation-mediated relaxation of chromatin structure (Figure 1) [3,9,18,22,23]. This is countered by competing interactions between Max, four structurally related bHLH-ZIP proteins (Mxd1-4) and the more distant bHLH-ZIP relatives Mnt and Mga. Collectively known as the “Mxd family”, these six factors are expressed in tissue-specific and developmentally-dependent ways [9,20]. Mxd-Max heterodimers displace Myc-Max heterodimers from their E box sites and in doing so reverse the previous Myc-dependent histone modifications by actively recruiting histone deacetylases, which re-compact chromatin and silence gene expression [9,18,22]. Thus, the transcriptional control of positively-regulated target genes is dependent upon the balance between these Mxd factors and Myc and their competition for Max [9]. In contrast, as many as 40–50% of Myc target genes are negatively regulated via very different mechanisms involving the inhibitory interaction of Myc monomers or Myc-Max heterodimers with positively-acting transcription factors such as Miz1/ZBTB7 and Sp1 or Sp3 [9,43,44]. These factors display their own binding site preferences to initiator (Inr) elements in the case of Miz1/ZBTB17 and to GC-rich elements “Sp1 elements” in the case of Sp1 and Sp3 [9,43,44].

b. Relevant target gene sets controlled by the Myc Network. With only a few notable exceptions, Myc is needed to maintain both normal and neoplastic cell proliferation, particularly in vitro [9,45,46,47,48,49,50]. It is therefore not surprising that cell cycle-related genes were some of the first to be identified as direct Myc targets [17,22,51]. Among the more prominent of the subsequently identified direct targets are those that broadly participate in “metabolism” and “translation” [9,47,51,52,53,54,55,56,57,58,59,60,61]. Subcategories of these rubrics include a myriad of genes whose products regulate key aspects of glycolysis, glutaminolysis, fatty acid metabolism, mitochondrial structure and function, ribosomal subunit biogenesis, rRNA and tRNA transcription and translation [9,47,55,56,57,58,59,60,61,62,63,64,65,66].

## 3. The Mlx Network and Its Control of Gene Expression

a. Mlx Network components. The bHLH-ZIP transcription factor members of the Mlx Network are structural and functional cousins of their Myc Network counterparts, and in some cases are identical (Figure 1) [9,20,28]. The Myc-like proteins, MondoA/MLXIP (hereafter MondoA) and ChREBP/MLXIPL (hereafter ChREBP), interact with the Max-like protein Mlx and bind as heterodimers to specific elements termed “carbohydrate response elements” (ChoREs), which are comprised of two imperfect E boxes (CAYGNGN_5_CNCRTG) separated by five nucleotides (Figure 1) [36,67,68,69,70]. However, recognition of true E boxes also occurs just as Myc-Max heterodimers can engage some ChoREs [9,38,70,71,72]. So-called “glucose-sensing modules” (GSMs) in the N-terminal domains of MondoA and ChREBP also bind glucose and glucose-6-phosphate (G6P), which are needed for the nuclear-to-cytoplasmic translocation of the heterodimers [9,31,32,33,34,35]. DNA binding is further influenced by intracellular pH and the levels of adenine nucleotides although the mechanisms underlying this are less clear [29,73,74]. Unlike Myc, which is ubiquitously expressed by and necessary for sustaining most proliferating cells, MondoA and ChREBP expression tends to be proliferation-independent and more tissue-restricted, with the former being most highly expressed in skeletal muscle and the latter predominating in liver and white adipose tissue [75,76,77]. Finally, unlike Max, whose two isoforms (p20 and p21) tend to be expressed at similar levels and localize to the nucleus, Mlx has three isoforms, the abundance of which varies among cell types, with only the longest γ isoform localizing to the nucleus [71].

As is true for the Myc Network, the Mlx Network also actively suppresses target genes, with the participating heterodimers being comprised of Mlx and three of the same repressive factors used by the Myc network, namely Mxd1, Mxd4 and Mnt (Figure 1). Together with shared DNA binding sites, this provides another direct means of cross-talk between the two Networks..

b. Relevant target gene sets controlled by the Mlx Network. Given the exceedingly high energy demands of proliferation and protein translation [40], it is probably not coincidental that they are coordinated by and linked to the ATP-generating processes of glycolysis and Oxphos via the Extended Myc Network. The fact that MondoA and ChREBP also tend to reside in proximity to cytoplasmic lipid droplets and that lipid depletion is an additional signal for their nuclear translocation further emphasizes the reliance on intracellular nutrients for the transcriptional activation of the pathway [78].

As mentioned above, Myc directly regulates hundreds to thousands of genes although their precise numbers and the magnitude of the regulation depend on cell type, stage of development and Myc levels [9,18,20,41,42,51]. This may also reflect the levels of Mxd members, which can vary widely for the same reasons (Figure 1) [9]. Target genes of the Mlx Network are fewer in number but show considerable overlap with those of the Myc Network [20,33,37,50,68,79,80]. Recent studies in *Myc−/−* (*Myc*KO), *Mlx−/−* (*Mlx*KO) and “double knockout” *Myc−/−* × *Mlx−/−* (mmDKO) murine embryonic fibroblasts (MEFs) and hepatocytes have indicated that, in addition to the gene sets mentioned above, those pertaining to aging, senescence and responses to and repair of DNA damage are also common targets [50,80].

## 4. Co-Regulation of Target Genes by the Extended Myc Network

a. Co-binding to shared target gene sites. The fact that Myc and Mlx Network members can cross-bind to one another’s DNA sites in some genes whereas in other cases, they bind separate E boxes and ChoREs provides additional opportunities for distinct types of co-regulation (Figure 2) [28,38,39,50,72,73,81]. It further implies that the control of both classes of genes may be quite dynamic, context-dependent and driven by the abundance and/or sub-cellular distribution of each Network’s components as well as the availability of certain metabolites. Examples where such situations might be particularly salient include tumors where high levels of Myc-Max might displace MondoA/ChREBP-Mlx heterodimers from their target genes, particularly when the latter’s binding sites are low-affinity or when the tumor micro-environment is deprived of glucose and/or highly acidic, as commonly occurs when tumor proliferation outpaces the vascular and nutrient supply [9,82] (Figure 2). Indeed, the loss of nutrient responsiveness due to the displacement of Mlx Network members by excessively high levels of Myc might explain why some tumors achieve such a situation to begin with. It might also occur during episodic proliferation in non-transformed cells when Myc is induced as cells exit the G_0_ state and enter S-phase. Conversely, Myc-Max binding to low-affinity E boxes or ChoREs might be displaced by MondoA/ChREBP-Mlx heterodimers as the latter translocate to the nucleus in response to changes in intracellular metabolites and/or pH or as Myc levels naturally decline during quiescence [28,29,30,31,32,33,34,35]. Mlx Network control over target gene expression might also be particularly pronounced in certain normal quiescent tissues where Myc levels are low but where Mlx Network members are high. Examples include but are not limited to skeletal muscle, liver, and adipose tissues where MondoA and ChREBP are constitutively expressed and where glucose and lipid metabolism are particularly pronounced and inter-connected [9,31,36,76]. The sharing of Mxd1, Mxd4 and Mnt by the Myc and Mlx Networks provides additional means by which these factors can directly communicate and “sense” one another’s environmental and metabolic status and the occupancy of distinct binding sites (Figure 1) [9,20,33,34,35,75,78,83].

One actual example of such transcriptional fine-tuning involves the *TXNIP* (thioredoxin-interacting protein) gene, which encodes a member of the α-arrestin family that is negatively regulated by Myc and positively regulated by MondoA and ChREBP (Figure 3) [72,84,85,86,87,88,89,90,91]. These opposing forms of control appear to result from competition between Myc-Max and MondoA/ChREBP-Mlx heterodimers for a pair of ChoREs in the *TXNIP* gene’s proximal promoter [87,88]. In addition to its involvement in maintaining intracellular redox balance, TXNIP also negatively regulates several important growth factor signaling pathways and glucose uptake, with the latter being achieved indirectly via TXNIP-regulated endocytosis of the *SLC2A1*- and *SLC2A4-*encoding Glut1 and Glut4 glucose transporters, respectively [32,84,85,89,90]. This effect on glucose uptake represents one arm of a feedback loop in which the nuclear translocation of MondoA-Mlx heterodimers (and probably ChREBP-Mlx heterodimers as well), is controlled by the previously mentioned interaction with glycolytic intermediates, which in turn is dictated by the rates of glucose uptake and glycolysis [91]. Completing this loop is Myc’s direct up-regulation of both *SLC2A1* and the *HK2* gene [53]. *HK2′*s encoded enzyme, hexokinase 2, catalyzes the first step in glycolysis that converts glucose to G6P [53]. Most other enzymes that mediate glycolysis are Myc-regulated as well with the gene encoding the liver and erythrocyte-specific isoform of pyruvate kinase (*PKLR)* being another that is positively co-regulated by ChREBP and Myc in ways that are variably responsive to glucose [9,53,63,91]. Thus, under conditions of rapid growth, high Myc levels function not only to facilitate growth but also to increase glycolysis while simultaneously suppressing an important factor, namely TXNIP, that limits glucose uptake.

## 5. The Myc and Mlx Networks Oversee Common Functions and Overlapping Gene Sets

a. The Extended Myc Network’s roles in normal hepatocyte proliferation. The strong but transient induction of Myc in the normal liver in response to proliferative stimuli was first observed over 35 years ago. These signals included partial hepatectomy (PH), carbon tetrachloride- or phenobarbtol-induced injury and treatment with the peroxisome proliferator-activated receptor α (PPARα) agonist Wy-14,643 [92,93]. In all cases, Myc appeared within minutes-hours and normalized well before full recovery from the injury. Thus, Myc induction was not only among the earliest molecular events observed but also remarkably independent of the nature of the inciting proliferative signal.

Determining whether Myc induction was actually necessary for proliferation was directly addressed in vivo with a *LoxP/Cre* strain of mice that allowed for *Myc’*s hepatocyte-specific knockout (KO) [94]. By measuring the proliferative response to Wy-14,643, Qu et al. first concluded that there was little noticeable difference in the proliferative response of wild-type (WT) and KO hepatocytes following exposure to the drug [95].

The second approach, used by 3 different groups, involved the conditional excision of *Myc* with varying degrees of efficiency (approx. 50–90%) and then shortly thereafter measuring the regenerative response to PH over the ensuing 2–7 days. Baena et al. determined that Myc was necessary for regeneration in adult livers but not in neonatal livers whereas Li et al. and Sanders et al., both concluded that Myc was dispensable for liver regeneration in young adult mice [96,97,98].

PH, while long considered the “gold standard” by which liver regeneration is appraised, has several limitations and drawbacks [99]. First, post-PH regeneration is complete within 7–10 days, thereby precluding long-term periods of observation. If the contribution of Myc (or any other factor) to hepatic regeneration is relatively small, it might only be appreciated during extended periods of hepatocyte injury and proliferation such as those associated with steatosis, early stage cirrhosis and alcoholic and non-alcoholic hepatitis [100,101,102]. Second, large numbers of animals might be needed to achieve statistically meaningful results if the differences between the groups being compared are relatively small. Third, given that the liver is comprised of ~60% hepatocytes by cell number [103], replacement of the missing mass following standard two-thirds PH requires fewer than two hepatocyte divisions, which hardly represents a robust proliferative demand. Fourth, as much as 40% of the restored liver mass following PH is a result of hypertrophy of the residual hepatocytes rather than de novo replication [104,105], which further minimizes the actual proliferative contribution. Finally, the point at which hepatic regeneration is considered to be complete following PH is somewhat murky and the methods to assess this often differ. Indeed, each of the studies cited above relied on different end points and mostly indirect criteria to assess hepatocyte proliferation [95,96,97,98].

Reasoning that PH might be insufficiently sensitive to identify Myc-dependent effects on hepatocyte proliferation, we have instead relied upon a murine model of Type I hereditary tyrosinemia (HT) [106]. HT is an autosomal recessive disorder that, in humans, is caused by mutations in fumarylacetoacetate hydrolase (FAH), the final and rate-limiting enzyme of hepatic tyrosine catabolism (Figure 4) [107]. The inheritance of defective *FAH* genes leads to the gradual accumulation of toxic tyrosine breakdown products such as succinylacetate, progressive hepatocyte death and eventual hepatic failure. A *Fah* KO mouse model of HT fully recapitulates the human disease [106]. As is true for the human disease, the consequences of Fah loss can be ameliorated by inhibiting the upstream enzyme 4-hydrooxyphenylpyruvate dioxygenase (HPD) with the competitive inhibitor Nitisinone (NTBC), which avoids the accumulation of toxic tyrosine catabolites and the associated hepatocyte damage (Figure 4). A second way to treat the disease is with liver transplantation or, in the case of mice, with the intrasplenic administration of *Fah+/+* hepatocytes, which then migrate to the liver, gradually replace the recipient’s moribund *Fah−/−* hepatocytes and ultimately contribute as much as 80% to the recipient’s hepatocyte mass [106]. Rather than the short period of limited hepatocyte proliferation that occurs is response to PH, the FAH model allows for a 50–100-fold expansion of donor *Fah+/+* cells over several months. The model also allows for the delivery of two or more populations of *Fah+/+* hepatocytes to the same animal, thereby allowing competitive studies to be performed in a manner analogous to that long done with hematopoietic cells [108]. Assuming that the donor populations can be distinguished from one another and from recipient hepatocytes (usually with sensitive and highly quantitative qPCR-based strategies), comparing the relative amounts of each present at the beginning and end of the study allows for a direct determination of their relative proliferative fitness within the identical environment. Importantly, this assay also measures donor population expansion directly, thus avoiding the need to employ the less quantitative methods used with PH models [95,96,97,98].

The FAH model has been used to compare the repopulation potential of WT hepatocytes to those in which *Myc* had been knocked out peri-natally by Cre recombinase driven by an albumin promoter (*Myc*KO hepatocytes) [109]. Initial experiments performed in recipient *Fah−/−* animals receiving only single *Fah+/+* donor hepatocyte populations showed that the time to achieve Nitisinone independence, the total liver weights and the contribution of donor cells to recipient livers were the same in the WT and KO donor groups. Importantly, immuno-staining of tissue sections performed near the end of the observation period showed that Myc protein was expressed only in the regenerating FAH+ hepatic nodules originating from the expansion of WT hepatocytes. Thus the regenerative capacity of *Myc*KO hepatocytes was not due to the selective expansion of a minority population that had escaped *Myc* gene excision and retained a proliferative advantage. Repeating these experiments under competitive transplant conditions confirmed that WT and *Myc*KO hepatocytes were equally proficient at reconstituting the recipient livers [109]. These results provided the most direct and compelling evidence yet that Myc expression was dispensable for liver regeneration even when the replacement process occurred over an extended period of time.

In follow-up to the above studies, the contributions of the Myc and Mlx Networks to hepatic regeneration were compared. The study’s rationale was based on the above-discussed dispensability for Myc in hepatocyte proliferation [109], on the high level of ChREBP expression in liver [75,77] and on previous evidence for cross-talk between and overlapping transcriptional repertoire of the Myc and Mlx Networks (Figure 1) [9,20,28,38,73]. In this study, competitive transplants into *Fah−/−* mice were performed with mixed populations of WT and *Chrebp*KO hepatocytes or WT and *Myc*KO × *Chrebp*KO double KO” (mcDKO) hepatocytes [79]. In the first case, a donor cell population comprised of 61.5% *Chrebp*KO cells was reduced by more than half (28%) after hepatic reconstitution, thereby indicating their significant competitive disadvantage [79]. Even more striking results were obtained with mcDKO, which declined from 53% in the input donor cells to only 7% in the fully repopulated liver. These results indicated that the loss of ChREBP was associated with a significant impairment of hepatocyte repopulation. It also suggested that, whereas *Myc* loss alone was insufficient to inhibit repopulation [109], it nevertheless reinforced *Chrebp* loss to further compromise regeneration.

The above study raised two questions. First, despite hepatocytes expressing particularly high levels of ChREBP, they also express enough MondoA to ask whether it might partially rescue *Chrebp*KO phenotypes. Second, the conclusion that mcDKO hepatocytes were more compromised than *Chrebp*KO hepatocytes was only inferred given that it was based on separate competitive transplants that did not directly pit the two KO populations against one another. Therefore, to address both questions in the same study, Wang et al. examined the consequences of genetically ablating *Mlx*, which allows for the functional inactivation of both ChREBP and MondoA (Figure 1) [37]. They then performed competitive repopulation studies with WT and MlxKO hepatocytes and with WT and *Mlx*KO x *Myc*KO double knockout (mmDKO) hepatocytes (Figure 1). In the first case, anticipating that *Mlx*KO hepatocytes would be even more replicatively challenged than *Chrebp*KO hepatocytes, the recipient mice were administered a donor population comprised of a 1:6 ratio of WT to *Mlx*KO hepatocytes. Despite this large donor cell numerical advantage, *Mlx*KO cells nonetheless eventually comprised only 4% of the recipient livers’ donor cells, strongly suggesting that MondoA also played an important role in supporting hepatocyte proliferation. Most likely as a result of the poor replicative capacity of *Mlx*KO cells, the extent of hepatic reconstitution by the donor population was also significantly lower than that seen in previous experiments, even when mice were maintained on Nitisinone for up to six months. An equally pronounced replicative defect was observed with mmDKO hepatocytes, which despite their being transplanted at a 10-fold excess relative to WT hepatocytes, eventually comprised only ~5% of the final donor cell population.

Finally, to compare the replicative capacities of *Mlx*KO and mmDKO hepatocytes more directly, competitive transplants were performed with a 1:1 ratio of the two populations [37]. After six months of Nitisinone treatment and an inability to the mice to be weaned from the drug, the average liver reconstitution by donor cells was only ~2%, which was in keeping with the previously observed poor replicative capacity of both donor populations. Nonetheless, this small donor population was, on average comprised of 95% *Mlx*KO cells. Therefore, despite their severe replicative disadvantage, *Mlx*KO hepatocytes still managed to outcompete mmDKO by a wide margin in a head-to-head comparison. In summary, these studies demonstrated that the relative repopulation potential of hepatocytes is unaffected by the loss of Myc, moderately compromised by the loss of ChREBP, even more compromised by the loss of Mlx and nearly abolished by the loss of both genes, which together inactivate the positive arm of the Myc Network and the entirety of the Mlx Network (Figure 1).

b. Mlx functions as a tumor suppressor. An unexpected finding observed in the livers of 36% of older *MlxKO* animals was the development of hepatic adenomas whose transcriptional profiles were distinct from those of both normal livers and hepatoblastomas (HBs) induced by the enforced expression of various mutant forms of β-catenin and yes-associated protein (YAP) [37,79,80,110]. In nearly all cases, these benign tumors, which occasionally contained small foci resembling hepatocellular carcinoma (HCC), did not re-express Mlx, indicating that they did not originate from a minority population of hepatocytes that had retained intact alleles of the gene. The previous study documenting the exceedingly slow growth of *Mlx*KO hepatocytes may have accounted for the occurrence of these tumors only in older mice. Together with previous findings of frequent *Mlx* copy number loss in a variety of human cancer types [9], these results provided additional evidence for Mlx as a suppressor of both malignant and benign tumors.

c. The Extended Myc Network’s role in fibroblast proliferation. Although *Myc−/−* embryos have long been known to die in mid-gestation, partially deleting *Myc* during embryogenesis allows for post-natal survival and results in the pups and their internal organs being smaller than normal due to the tissues containing fewer cells of otherwise normal size [46,111,112]. This is consistent with the finding that the proliferation rates of primary MEFs derived from these animals decline in parallel with Myc levels [46,50]. In a more recent study, the *Myc* gene was gradually inactivated in vitro over the course of 5–7 days after early passage primary MEFs had been generated [50]. These cells initially arrested in G_0_/G_1_ but then further consolidated in both G_0_/G_1_ and G_2_/M over the ensuing week [50]. It was suggested that this late-appearing G_2_/M-arrested population originated from cells that had been in S-phase at the end of the first week and then subsequently progressed into G_2_/M but were unable to exit and re-enter G_0_/G_1_ as Myc levels continued their decline [50].

An exception to the above findings was observed in an unusual strain of immortalized Rat fibroblasts in which *Myc* inactivation markedly suppressed but did not entirely inhibit proliferation [45]. Rather, these cells prolonged their doubling time to ~4–5 days versus 18–24 h for the *Myc+/+* cells from which they were derived. Normal-near normal doubling times could be restored by re-expressing Myc or other members of the Myc family (i.e., N-Myc and L-Myc) or any one of three different Myc target genes [113,114,115]. Thus, factors such as whether the cells being studied are transformed or otherwise immortalized or precisely when during development *Myc* was deleted appear to play significant roles in determining their reliance on Myc for maintaining proliferation.

The inactivation of *Mlx* in primary MEFs did not itself impair proliferation but did result in some unanticipated behaviors when combined with *Myc* knockout [50]. Not unexpectedly, these mmDKO cells initially growth-arrested in a manner closely resembling that of *Myc*KO MEFs while also displaying similar cell cycle profiles. However, between days 14 and 21, they spontaneously resumed proliferating at rates exceeding those of comparably aged WT MEFs. It was concluded that the permanent growth-arrest associated with Myc inactivation requires an intact Mlx Network and is consistent with Mlx’s previously proposed role as a tumor suppressor or negative regulator of growth [9,37,50].

That the permanent growth arrest of primary *Myc*KO MEFs could be reversed by co-inactivating *Mlx* prompted an investigation into whether a similar down-regulation of Mlx in *Myc−/−* rat fibroblasts might explain their continued, albeit markedly slowed, proliferation [45,47]. Whereas WT and *Myc−*/*−* cells expressed similar levels of the truncated α and β isoforms of Mlx, both of which localize to the cytoplasm because they lack GSMs at their N-termini [50], the latter markedly down-regulated the full-length γ-isoform, which does localize to the nucleus and is thus likely to be the transcriptionally active isoform [50,68,70]; 17 of the 33 different cancer types in The Cancer Genome Atlas also demonstrated significant direct correlations between Myc and Mlx transcript levels [50]. Collectively, these findings suggest that, while regulating certain common genes and phenotypes, the Mlx Network may also play a very specific role in opposing Myc Network proliferative signaling.

Another unexpected discovery was made when the above MEFs were first immortalized by SV40 T antigen prior to the knockout of *Myc* or *Myc+Mlx* [50,116]. Unlike primary *Myc*KO and mmDKO cells, which growth-arrested immediately upon target gene(s) excision, the immortalized KO cells continued to proliferate, albeit at somewhat slower rates. These studies indicated that the proliferative defects of *Myc*KO MEFs can also be rescued by inactivating one or perhaps both of the tumor suppressors that are targeted by T antigen, namely p53 and Rb [115]. The generalizability of these findings, the specific role(s) played by each factor and whether both p53 and Rb must be inactivated remains to be determined for other cell types. However, previous studies have shown that the *TFIIB* gene, which encodes a key factor involved in the RNA polymerase III-specific transcription of tRNA and 5S ribosomal genes and which is necessary for maintaining proliferation, is repressed by p53 and Rb and induced by Myc [58]. Similar p53- and Rb-mediated repression and Myc-mediated induction have been described for the telomerase reverse transcriptase (*TERT)* gene, which is necessary for imparting and maintaining the immortalized state in some cell types [117,118,119,120,121]. A clue that the Myc-independent proliferation of SV40 T antigen-immortalized MEFs and mmDKO primary MEFs might be driven by the same underlying mechanism was provided by comparing the RNAseq profiles of primary *Myc*KO and mmDKO MEFs [50]. Gene set enrichment analysis (GSEA) of the differentially expressed transcripts from these two cell lines revealed that the top categories contained multiple genes enriched for down-stream targets of p53 and Rb and those associated with the G_2_/M block. Each of these gene sets was enriched in ways that were consistent with the resumption of proliferation by the DKO cells [50].

Day 7–14 *Myc*KO and mmDKO primary MEFs, and to a lesser extent *Mlx*KO MEFs, also displayed a number of properties that were indicative of premature senescence and/or aging [122,123,124]. In addition to growth-arrest, these include a flattened morphology, increased lysosomal content and glucose uptake, higher senescence-associated β-galactosidase activity, the accumulation of neutral lipid, reduced rates of protein synthesis, impaired mitochondrial function and increased production of mitochondrial-derived reactive oxygen species (ROS). These properties were subsequently lost or at least markedly attenuated in primary mmDKO cells that had resumed proliferating. They also tended to be lost in SV40 T antigen-immortalized cells.

d. The Myc and Mlx Networks regulate common as well as distinct metabolic processes. The Myc and Mlx Networks regulate several metabolic pathways cooperatively, most notably those involving glycolysis and mitochondrial structure and function [9,18,28,32,33,34,37,47,50,55,56,57,61,62,63,73,75,78,79,80]. Studies in which various components of the two Networks have been knocked out have helped to illustrate the overlap between the two Networks as well as their individual functions [9].

Conspicuous abnormalities in glycolysis and mitochondrial structure and function were first documented in the previously mentioned, *Myc−/−* rat fibroblasts [45,47,61,62,63]. These cells contained small, atrophic mitochondria with a paucity of cristae and a ~10-fold lower membrane potential [47,61,62]. They also displayed much lower basal rates of glycolysis and Oxphos, a highly attenuated spare respiratory capacity and significantly reduced levels of glycolysis- and mitochondrial-related transcripts [47,63,64]. Collectively, these features undoubtedly contributed to the cells’ also having a three-fold lower ATP content [47]. Many genes that encode mitochondrial proteins are direct Myc targets and their upregulation by ectopic Myc expression is likely further enhanced by Transcription Factor A- mitochondrial (TFAM) an important mitochondrial DNA replication factor whose gene is also a direct Myc target [37,61,63,79,80].

Non-denaturing blue native gel electrophoresis (BNGE) and enzymatic assays of electron transport chain complex (ETC) function in *Myc−/−* fibroblasts showed up to five-fold reduced amounts and/or activities of Complexes I, II, III, V and of ETC supercomplexes comprised of varying stoichiometries of Complexes I, III and IV [47,125]. These cells also showed half the normal level of mitochondrial fusion along with dysregulated expression of several proteins that promote both fusion (Mfn1, Mfn2, Opa) and fission (Drp1, Fis1) [47,126]. Because continuous fusion is necessary to maintain maximal ETC efficiency, it was proposed that preventing or delaying the rejuvenation of old and functionally debilitated mitochondria with younger and healthier ones via fusion would amplify pre-existing bio-energetic abnormalities [127,128,129]. Re-expressing Myc corrected the above defects to varying degrees, with supraphysiologic levels of Myc increasing basal levels of glycolysis and Oxphos by more than two-fold relative to *Myc+/+* cells [47,61,64]. The above findings were extended to A549 human lung cancer cells in which doxycycline-inducible shRNA-mediated knockdown of endogenous Myc caused immediate proliferative arrest, a flattened morphology resembling that of *Myc−/− fibroblasts*, a two-fold reduction in mitochondrial mass, a ten-fold reduction in basal Oxphos and the complete failure to up-regulate Oxphos the usual two-three-fold in response to the Oxphos uncoupler FCCP and the glycolysis inhibitor 2-deoxyglucose [47].

Further consistent with the above studies were the results obtained when the mitochondrial fission protein Drp1 was constitutively expressed in immortalized but otherwise normal rat fibroblasts [126]. In vitro, purified Drp1 forms oligomeric rings around the constricting regions of mitochondria as they undergo fission; in doing so these rings promote further contraction and eventual scission [130]. Drp1 also helps to maintain the size, shape and cellular distribution of mitochondria and the remodeling of their cristae [130,131,132,133,134,135]. Drp1-overexpressing fibroblasts acquired a number of features that recalled those of *Myc−*/*−* rat fibroblasts [47] These included greatly reduced mitochondrial size and connectivity; a paucity of cristae; structural and functional deficits in ETC Complexes I and V; loss of Oxphos; elevated mitochondrial-derived ROS and chronic phosphorylation (activation) of AMP-activated protein kinase (AMPK), which is a general sensor of the ATP-depleted state that tends to remain active until ATP levels are normalized [126,136]. In addition to high rates of mitochondrial fission, comparably elevated rates of fusion were also observed that were interpreted as representing an unsuccessful attempt to restore a normal fission-fusion balance and ATP levels. An accompanying resistance to both serum-dependent and serum-independent apoptotic stimuli was shown to be directly attributable to these mitochondrial defects since normal apoptotic sensitivities could be restored by pharmacologically restoring ATP levels with AICAR (5-Aminoimidazole-4-carboxamide ribonucleotide). It was proposed that the ATP-depleted state that is frequently observed in tumors as a result of poor vascularization and micro-environmental nutrient depletion might actually be beneficial by allowing cells to circumvent the pro-apoptotic proclivities that can limit Myc over-expression and its associated oncogenic drive [126,137].

A wider role for Myc in maintaining ATP levels and the bio-energetic consequences of being unable to do so was subsequently extended to three different human cancer cell lines treated with nine different small molecule Myc inhibitors with four different mechanisms of action [49]. Within 16 h of treatment with these compounds, each cell line showed an abrupt proliferative arrest, a marked loss of Myc protein expression, up to 95% reductions in ATP levels and AMPK activation. In response to these ATP-depleting Myc inhibitors, the HL60 promyelocytic leukemia cell line also underwent variable degrees of myelo-monocytic differentiation, which in some cases matched the levels achieved in response to the classic and potent differentiating agents dimethylsulfoxide (DMSO) and the phorbol ester 12-O-tetradecanoylphorbol-13-acetate (TPA) [49].

Because Myc over-expression has long been known to block differentiation in a number of settings and cell types [138,139,140,141] it was speculated in the above study that merely depleting ATP might be sufficient to induce differentiation in a Myc-independent manner [49]. Indeed, treatment with either metformin or oligomycin A, which block ETC complexes I and V, respectively [47,142], caused rapid declines in ATP levels, activated AMPK and induced a degree of myelomonocytic differentiation that exceeded that achieved with DMSO, TPA or any of the small molecule Myc inhibitors. Moreover, Myc protein continued to be expressed at levels equaling or even modestly exceeding those of untreated cells. Collectively, these results strongly suggested a link between metabolism and differentiation and that a major means of blocking differentiation in response to Myc over-expression (at least in HL60 cells) involves its ability to sustain ATP levels.

A seemingly universal finding in the above cell lines and others following the inhibition or inactivation of Myc (or N-Myc in the case of neuroblastoma cell lines) was an increased rate of fatty acid β-oxidation (FAO) and the cytoplasmic accumulation of neutral lipid [47,49,56,57,143,144,145]. Given that this tended to be associated with AMPK activation [49], it was speculated that it represented an unsuccessful attempt to restore ATP levels by switching from glucose to lipid as a secondary and more efficient energy source, as typically happens during starvation or when glucose availability is otherwise restricted [57,63,146,147]. Given that mitochondrial function itself was compromised, these increases in FAO were unable to compensate for the ATP deficit. The increased uptake of fatty acids therefore exceeded the oxidative capacity of the defective mitochondria and the difference was stored as neutral lipid in lipid droplets.

Primary growth-arrested *Myc*KO MEFs also showed significant metabolic disturbances although they were somewhat different and less pronounced than those encountered in iummortalized Rat *Myc−/−* fibroblasts [47,50]. For example, rather than being decreased, the total mitochondrial mass of the former was actually increased about two-fold relative to that of WT MEFs. Baseline mitochondrial oxygen consumption rates (OCRs), responses to pyruvate, malate and succinate and the overall function of Complexes I and II, while initially appearing indistinguishable from those of WT MEFs, were in fact reduced by half after taking into account this difference in mitochondrial mass. Further consistent with mitochondrial dysfunction was the demonstration that *Myc*KO cells also accumulated neutral lipid and produced higher levels of ROS, most of which appeared to be of mitochondrial origin [50]. Collectively, these findings indicated that mitochondrial compromise in response to genetic or pharmacologic Myc inhibition is variable in its severity but nearly always involves the loss of ETC function and/or structure, reduced ATP stores, increased FAO, neutral lipid accumulation and elevated ROS that likely originates from faulty electron transport [47,62].

Consistent with the finding that primary *Mlx*KO MEFs proliferated at rates comparable to those of their WT counterparts, few if any changes indicative of mitochondrial dysfunction were observed [50]. Although the total mitochondrial mass of these cells was slightly less than that of WT cells no significant differences in OCRs were observed in response to TCA cycle substrates, in the production of mitochondrial-derived ROS or in neutral lipid accumulation. Finally, mmDKO MEFs had profiles similar to those of *Myc*KO MEFs indicating that the loss of the *Mlx* Network did not further impact the deficits arising in response to *Myc* loss.

The consequences of *Myc* inactivation on mitochondrial structure and function in the liver were examined in the same study that had demonstrated no effect on hepatocyte repopulation [109]. BNGE performed on partially purified liver mitochondria from *Myc*KO mice failed to show any of the marked differences in ETC complex structure and function previously observed in fibroblasts [47]. A mass spectrometry-based quantification of ~30% of the mitochondrial proteome, including all 93 subunits of Complexes I–V, also failed to demonstrate any differences between WT and *Myc*KO livers. Despite these indistinguishable profiles, the OCRs of *Myc*KO liver mitochondria in response to ADP, pyruvate, malate and succinate were diminished relative to the WT mitochondria although Complex V activity, as measured by the in situ assessment of ATPase function was increased. The significance of this latter finding is unclear although it conceivably represents a compensatory change intended to normalize the consequences of the Complex I and Complex II functional deficits. Thus, despite the lack of any demonstrable structural abnormalities resembling those of *Myc*KO fibroblasts, *Myc*KO livers nevertheless demonstrated similar, albeit less severe, functional impairments. This argued that, in the context of hepatocyte transplantation studies (Section 5.a) [109], the energy-generating defects of these cells were not severe enough to compromise their otherwise relatively slow, normal and controlled proliferative response. On the other hand, the abnormalities in Oxphos may have contributed to the slower growth rates of HBs in these livers by virtue of being unable to provide sufficient ATP to support the much more rapid and deregulated rates of neoplastic growth [79]. The point at which the energy demands of replication outstrip the ability to provide adequate amounts of ATP from Oxphos and glycolysis may therefore lie somewhere between the highly controlled and relatively slow proliferation of normal hepatocytes and the uncontrolled rapid proliferation of tumors. Importantly, the fact that Myc’s absence did not impair the induction of tumorigenesis proved that, at least in this restricted context, Myc’s role is to maximize tumor growth but not to initiate it.

Further evidence of subtle functional defects in *Myc*KO liver mitochondria was obtained by noting that the livers of young *Myc*KO mouse contained higher levels of triglycerides as well as larger and more numerous neutral lipid droplets than seen in the livers of similarly aged WT mice [109]. The differences between these two groups became even more pronounced following transplantation of their respective hepatocytes into *Fah-/-* recipients, with much of the neutral lipid of *Myc*KO hepatocytes now residing in larger droplets and with significant extracellular deposits. Taken together, the findings in fibroblasts, livers and other cells in which Myc or N-Myc have been inactivated either genetically or pharmacologically indicate the most consistent metabolic defect(s) to be the accumulation of excess lipid. As already discussed above, this is presumed to be the consequence of an exaggerated uptake of exogenous fatty acids that outpaces their ability to be catabolized into energy-generating substrates in functionally defective mitochondria [47,49,79,109,144,145].

Two more in vivo studies have examined the cooperative cross-talk between the Myc and Mlx Networks in maintaining metabolic balance in the liver [37,79]. In the first study, the previously described *Myc*KO, *Chreb*KO and mcDKO livers were compared [79]. The hepatic neutral lipid and triglyceride content of *Chrebp*KO and mcDKO livers was more elevated than that of *Myc*KO livers which, as noted previously, was already higher than that of WT mice [79,80] This suggested that the Mlx Network played a more important role in controlling hepatic lipid metabolism (or perhaps in controlling mitochondrial structure and function) than did the Myc Network. In the second study, additional groups of mice were examined that included the *Mlx*KO and mmDKO groups [37]. Older mice (~14–60 months) from all the above KO groups were also compared to determine how age impacted the accumulation of lipid. All KO mice showed comparable levels of steatosis with hepatic triglyceride levels that were ~six-eight-fold higher than those of WT control mice of identical age. Together with the previously mentioned study in much younger animals, these findings suggested that neutral lipid accumulates earlier in *Mlx*KO and mmDKO mice than it does in *Myc*KO, *Chrebp*KO or mcDKO mice. Eventually, however, all KO groups accumulated about the same abnormally high lipid content with little additional accrual and thereby suggesting that this represented an upper limit of hepatic lipid capacity. A group of 163 liver transcripts, selected because of their strong association with non-alcoholic fatty liver disease also showed a progressive enrichment as the Extended Myc Network was progressively dismantled [37].

e. The overlap of target gene expression. Global transcriptional profiling performed on primary *Myc*KO, *Mlx*KO and mmDKO MEFs 10 days after initiating knockout of the respective genes [50] confirmed previous reports that the target gene repertoire of the Mlx Network was smaller than that of the Myc Network [9,28,37,79]. Among the >6300 transcripts collectively dysregulated by the three KO lines, the number in *Mlx*KO cells was less than one-fourth as large as that of *Myc*KO cells, with 61.8% of Mlx-regulated transcripts being co-regulated by Myc and 14.5% of Myc-regulated transcripts being co-regulated by Mlx. ChIP-seq results from the ENCODE data base [148] indicated that as many as 64.1% and 29.6% of the dysregulated transcripts in *Myc*KO and *Mlx*KO MEFs, respectively, were direct targets of these factors, with 22% of these directly binding both Myc and Mlx. In accordance with the model shown in Figure 2, it was imagined that some of the target gene-associated E boxes and ChoREs identified in this screen might be susceptible to binding by factors from both the Myc and Mlx Networks depending on the various circumstances and conditions discussed above.

To identify functionally related categories of genes under the control of the above Networks, both directed and unbiased GSEA were performed using the EnrichR tool kit, which contains a large number of independent data bases, including those from the KEGG, mSigDB C2 and Mitoproteome repositories [37]. Seven major groups of differentially enriched gene sets were identified. One such group, comprised of 39 individual gene sets, was related to mitochondrial structure and function. Although 28 of these gene sets were significantly enriched in *Myc*KO MEFs and 31 were enriched in mmDKOs MEFs, none were enriched in *Mlx*KO MEFs. This was very different from the case in livers where many of the same mitochondria-related genes that had been identified as being mostly down-regulated in *Myc*KO livers were more down-regulated in *Chrebp*KO, *Mlx*KO livers and even more so in mmDKO livers [3,9,79]. However, the findings in MEFs were consistent with the much milder mitochondrial consequences observed in both primary and immortalized *Mlx*KO cells [50]. Another of the top enriched transcript groups was comprised of 43 gene sets pertaining to cholesterol metabolism with 39 being enriched in *Myc*KO MEFs relative to WT, 30 being enriched in mmDKO MEFs and 13 being enriched in *Mlx*KO MEFs [149,150]. Collectively, these findings extended the defects in lipid metabolism into the realm of cholesterol biosynthesis and indicated a more restricted role for the Mlx Network in this process.

As mentioned above, primary *Myc*KO and mmDKO MEFs also displayed numerous features that were consistent with premature aging and senescence [50]. Thus it was not surprising that the additional top enriched gene sets from these groups included those involved in aging (34 gene sets) and senescence (13 gene sets) as well as ribosome structure and function (25 gene sets) and cell cycle regulation (>50 gene sets) [50]. A final large group comprised of 49 gene sets representing DNA damage recognition and repair processes, was particularly enriched in *Myc*KO and mmDKO MEFs (49 and 46 gene sets, respectively) with ten of the sets also being enriched in *Mlx*KO MEFs [37]. Given that many inherited and acquired defects in these pathways are associated with premature aging and senescence, it was proposed that the global dysregulation of these multiple DNA damage recognition/repair pathways might also accelerate the premature aging and senescence phenotypes [151,152,153,154].

Comparative RNAseq has also been performed on WT, *Mlx*KO and mmDKO livers and analyzed with gene sets previously shown to comprised of transcripts encoded by direct Myc or MondoA/ChREBP target genes. A number of differentially expressed transcripts overlapped, particularly in the case mmDKO livers whose dysregulated transcript repertoire was the largest. This again pointed to at least additive if not synergistic cooperation between the Myc and Mlx Networks in regulating transcripts previously thought to be under the purview of only one Network (Figure 2) [20,28,32,79]. The dysregulated gene repertoire in *Mlx*KO livers was also larger than that of previously described *Chrebp*KO livers, once again attesting to the redundant function of MondoA and ChREBP [28,32,75,76,78].

Having previously examined the outcomes of altered gene expression profiles in *Myc*KO and *Chrebp*KO livers, they could now be compared with those that accompanied progressive inactivation of the Extended Myc Network [37]. This demonstrated degrees of transcriptional dysregulation that became deeper and more encompassing as Extended Myc Network member inactivation progressed step-wise in the following manner: *Myc*KO < *Chrebp*KO < mcDKO < *Mlx*KO < mmDKO. Ingenuity pathway analysis profiling of the differentially expressed transcripts from the latter two groups showed (as anticipated) that the top functional categories of transcripts encoded proteins involved in mitochondrial and ribosomal structure and function, thus confirming previous findings and those of the MEF studies described above [37,50,55,80]. Searches of the ENCODE chromatin immunoprecipitation-sequence (ChIP-seq) data base identified 4152 genes whose proximal promoter regions (defined as within +/*−* 2.5 kb of the transcriptional start sites) bound Myc, 748 that bound Mlx and 2433 that bound both Myc and Mlx [37,148]. As a group, Mlx-bound genes were more than twice as likely to also bind Myc as Myc-bound genes were to also bind Mlx (76% vs. 37%). Further contributing to the notion that a significant fraction of this overlapping gene set co-bound members of both Networks at shared sites was the finding that ~75% of the Myc and Mlx binding peaks mapped to within 170 bp of one another and that half mapped to within 65 bp. The close proximity of many of these peaks thus indicated that the ChIP-seq results alone were insufficiently sensitive to determine whether Myc and Mlx were binding to the same sites or to distinct sites close to one another. This issue was resolved by examining the actual sequences flanking the non-resolvable Myc and Mlx binding peaks. 42.6% of these sites contained only consensus E box elements and 24.9% contained only consensus ChoREs. Thus a substantial fraction of E boxes and ChoREs appear capable of binding members of the companion Network [9,38]. Sites containing neither obvious E boxes nor ChoREs were assumed to be recognized by Myc and/or Mlx Network members via non-consensus elements or (more likely) to bind indirectly as a result of associating with other DNA binding factors such as Miz1/Zbtb17 and Sp1 [21,43,44].

A separate analysis of the above target genes’ proximal promoters indicated that the E boxes and ChoREs contained within them were not randomly distributed relative to one another. Rather, two general types of binding sites were observed. The first contained only a single consensus element, indicating that both Myc and Mlx likely shared one another’s binding sites as mentioned above. In the second case, multiple tightly clustered E boxes and ChoREs were observed. This indicated that many of the previously observed ChIP peaks likely represented integrated signals from multiple sites that either co-bound Myc and Mlx Network members or bound them in such close proximity that they could not be resolved [37].

The functional categories of genes identified by these binding differences were also quite distinct. For example, genes bound exclusively by Myc were more likely to be involved in cell cycle regulation, growth factor responses and Oxphos whereas Mlx-bound genes were more likely to be involved in FAO, the TCA cycle and xenobiotic metabolism. However, these functional categories overlapped considerably and particularly so in the case of those involving ribosome structure and function and translation [37]. Moreover, the testable model depicted in Figure 2 suggested that these gene sets might not necessarily be fixed with regard to factors responsible for their expression; rather, they might be subject to regulation by one or both Networks based on different environmental, metabolic and proliferative cues, tissue types, stage of differentiation and the prevailing levels of Myc and Mlx Network members.

In cells expressing high or otherwise “pathological” levels of Myc, most if not all of the genes that encode glycolytic enzymes, as well as *LDHA,* which encodes the A isoform of lactate dehydrogenase, are up-regulated and thus collectively contribute to the Warburg effect via a common mechanism [9,53,64,65,81,155,156]. This is likely to be reinforced indirectly as well due to Myc’s induction of the *PDK1* gene, which encodes a kinase (pyruvate dehydrogenase kinase 1) that phosphorylates mitochondrial pyruvate dehydrogenase (PDH) at three sites, inhibits its activity and helps to redirect pyruvate’s to lactate instead of acetyl coenzyme A and thereby providing the electron acceptor NAD+ necessary for sustaining the more proximal steps of glycolysis [157,158]. The gradual dismantling of the Extended Myc Network in the liver, which expresses low levels of Myc and high levels of ChREBP and Mlx, was associated with the progressive down-regulation of only 3 glycolysis-related transcripts all of which are rate-limiting. These included liver-type pyruvate kinase (Pklr), liver-type phosphofructokinase (Pfkl) and the solute carrier gene *Slc2a2*, which encodes the major hepatic glucose transporter Glut2 [159]. Support for the idea that *Pklr* expression is directly regulated by both the Myc and Mlx Networks had been previously provided by work performed in rat insulinoma cells which showed the *Pklr* gene’s proximal promoter to be associated with paused RNA Pol II that accumulated around the transcriptional start site [72,81,160,161]. Providing glucose led to the C-terminal phosphorylation of Pol II and its read-through into the gene’s coding region, the hyperacetylation of neighboring histones H3 and H4 and the recruitment of ChREBP to a consensus ChoRE in the proximal promoter. Myc rapidly associated with the promoter as well indicating that its binding was dependent upon ChREBP co-binding. While Myc binding was distributed more broadly than ChREBP binding, the peaks for the two factors coincided. As this region contained no consensus E boxes, the simplest explanation for these observations was that Myc and ChREBP compete for the same ChoRE element and coordinately and cooperatively overseeing glucose-dependent transcription. This interpretation was further reinforced by showing that *Pklr*’s induction in response to glucose could be abrogated by the highly specific Myc inhibitor 10058-F4 [72,162]. The co-regulation of *Pklr* expression by Myc and ChREBP is reminiscent of the regulation of *TXNIP* described above [87,88]. Although the direct regulation of *Slc2a2* and *Pfkl* by either the Myc or Mlx Networks has not been reported, we have identified the direct binding of Myc to the proximal promoters of these genes in HepG2 HB cells using data from the ENCODE consortium as well as binding of Mlx to the promoter region of the latter gene [148], Wang and Prochownik, unpublished results. Others have also shown that the *PFKLR* gene could be up-regulated in HCC cells in response to Myc overexpression [163].

In summary, substantial regulatory overlap exists among genes that are subject to direct control by the two components of the Extended Myc Network (Figure 1). Those directly supervised by the Myc Network appear to be as much as three-four times more numerous than those regulated by the Mlx Network. Accordingly, the population of Mlx target genes that co-bind Myc appears to be about four times larger than the population of Myc targets that co-bind Mlx [37]. However, such numbers should be considered only as snap shots of events occurring at a particular time in single cell lines or tissues and are undoubtedly subject to change according to the relative and absolute amounts of each factor and various environmental and metabolic conditions. Genes previously considered as being regulated only by the Mlx Network and containing only ChoREs in their promoters may become transiently or permanently subject to direct Myc regulation during periods of high expression such as occur during normal proliferation or tumor-associated dysregulation. Likewise, genes under the exclusive purview of Myc in proliferating cells may now become more Mlx Network dependent during quiescence and/or during exposure to a glucose-rich environment [9]. Such altered regulation may involve entire functionally related gene sets and thus provide the means of achieving the coordinated responses necessary to alter complex multi-component cellular processes such as glycolysis, mitochondrial structure and function, translation and proliferation.

## 6. Conclusions and Future Directions

The more than two decade period during which the individual components of the Extended Myc Network have been identified and their functional frameworks and cross-talk contextualized has yielded a model that is both simple and elegant in its design (Figure 1) [9,18,20,21,22,28,33,51,140]. More recently, the means by which each individual Network participates in the regulation of target gene expression and the inherent similarities and differences in this control under different environmental conditions and in response to nutrients have been clarified by numerous in vitro and in vivo studies (Figure 2 and Figure 3). Among the generalizations that have emerged from this work is that the requirements for each Network’s members can vary markedly in ways that reflect the cells or tissue in which they are being studied, the conditions to which they are subjected and the stage of development. Such examples include the absolute requirement for Myc to support the proliferation of both normal and neoplastic cells under some circumstances while at the same time being entirely dispensable in other normal tissue such as hepatocytes during controlled regeneration [9,46,48,49,80,95,96,97,98]. Indeed, even within the same tissue and under similar circumstances, the role for Myc can differ widely. For example, Myc dysregulation is both sufficient to induce aggressive HCCs and necessary to maintain them [16,55] whereas in the context of HB pathogenesis mediated by the enforced expression of mutant forms of β-catenin and YAP, the absence of Myc only slows tumor growth without impacting its induction [80].

Despite the depth and breadth of our current understanding, several major questions remain to be more fully resolved. Chief among these are the precise conditions under which Myc Network target genes become fully subject to regulation by the Mlx Network and vice versa. It is likely that changes in such target gene preference arise either as a result of members from both Networks binding to their individual consensus binding sites in close proximity to one another as well as to competition for the same binding sites. Do these represent functionally distinct gene sets? Elegant work has demonstrated the existence of a large complex of factors that associate with and are in turn regulated by Myc following its binding to promoter/enhancer regions of its target genes and that are necessary for transcriptional activation [3,23,164,165]. In contrast, much less is known concerning the relationship between these transcriptional co-factors and those recruited by MondoA and ChREBP, although histone acetylases are known to be involved [33]. How is gene expression fine-tuned in response to what are likely to be differences in the composition of these complexes and how does the proximity of the Myc and Mlx Network member binding alter these both qualitatively and quantitatively?

Another area for future exploration concerns the role(s) for the Mlx Network in modulating both normal and tumor-associated immune responses, particularly in the case of T cell activation where Myc has been deemed essential in up-regulating the bursts in glycolytic and glutaminolytic activities that accompany and are required for this process [166,167,168]. In contrast, little if any information is available concerning the degree to which lymphocyte activation relies upon the Mlx Network, either independent of or in cooperation with the Myc Network [169,170]. Studies that capitalize upon the inactivation of *Chrebp*, *Mondoa*, *Mlx* and/or other members of the Network might also be useful in determining the degree to which lymphoid activation depends upon the extracellular nutrient milieu to achieve the ideal degree of activation.

Might the Myc and Mlx Networks “communicate” by virtue of microRNAs (miRNAs) whose transcriptional regulation by one Network might post-transcriptionally regulate members of the other Network? Although Myc is known to induce the expression of many miRNAs and Myc mRNA is itself a miRNA target, little information exists regarding miRNAs in the regulation of or by any members of the Mlx Network [171,172,173,174].

Finally, as clinically relevant pharmacologic targeting of Myc slowly becomes an increasingly realistic goal using agents that inhibit the formation of Myc-Max heterodimers, their binding to DNA or their activation of RNA Pol II-mediated transcription, it is reasonable to ask how such drugs might affect binding by Mlx Network members [49,144,162,175,176,177]. For example, might these drugs alter the propensity of sites previously occupied by Myc to now bind Mlx Network members (Figure 2)? Might the down-regulation of these genes not be as pronounced as expected if they now bind MondoA and/or ChREBP and might they now become subject to nutrient control? Finally, given the evidence for Mlx’s role as a suppressor of both normal and neoplastic proliferation [9,37,50], is it possible that increasing its abundance or activity in tumors might cooperate with Myc inhibitors by independently suppressing proliferating in much the same way that p53 does when its normal activity is restored [178]? The next several years should provide answers to these questions while providing new ones for our consideration.

## Figures and Tables

**Figure 1 cells-11-03974-f001:**
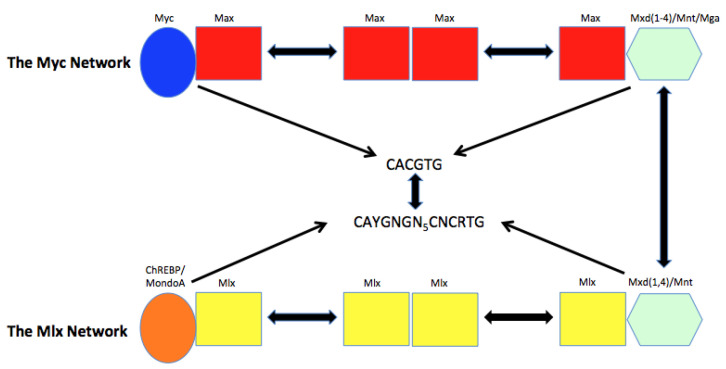
**The Myc and Mlx Networks.** Inactive as a monomer and unable to homodimerize under physiologic conditions, Myc binds to target gene consensus E-boxes (CACGTG) only after heterodimerizing with Max [9,22]. This leads to the de novo acetylation and methylation of specific histone residues, the relaxing of chromatin structure and target gene transcriptional activation following the Myc-dependent recruitment of scores of transcriptional cofactors [9,23,24,25]. Although Max homodimerizes and binds DNA well in its unmodified form, its ability to do so within cells is low due to inhibitory casein kinase II-mediated phosphorylation [26,27]. Transcriptional silencing is mediated by six “Mxd proteins” (Mxd [1,2,3,4], Mnt and Mga), which compete with Myc for Max, displace Myc-Max heterodimers from E-boxes, recruit histone deacetylases and reverse the above-described histone modifications [9,18,23]. In the Mlx Network, the Myc-like factors ChREBP and MondoA homodimerize poorly but heterodimerize well with the Max-like factor Mlx. However, they enter the nucleus and engage E box-related ChoRE sites (CAYGNGN_5_CNCRTG) only after binding glucose, glucose-6-phosphate, fructose-2,6-bisphosphate, lactate or adenosine [9,28,29,30,31,32,33,34,35,36]. The reversal of ChREBP/MondoA target gene activation is mediated by heterodimers between Mlx and Mxd1, Mxd4 and Mnt, thus allowing communication with the Myc Network. Further cross-talk derives from the fact that Myc and Mlx Network members may bind one another’s target genes under certain circumstances [9,28,37].

**Figure 2 cells-11-03974-f002:**
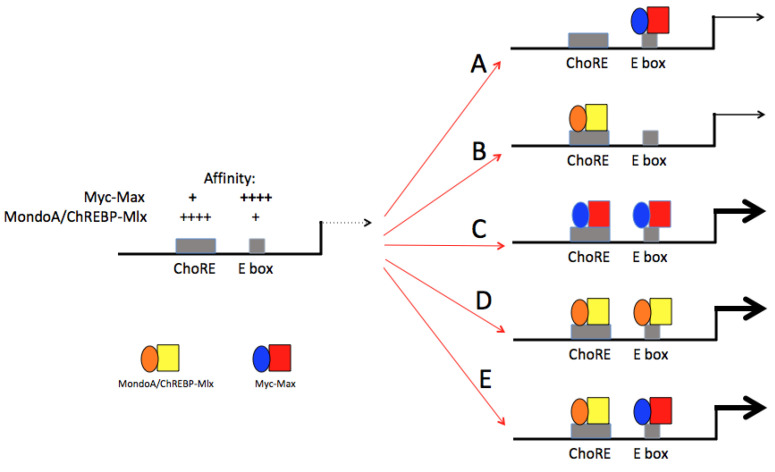
**Co-regulation of common target genes by Myc and Mlx Network members.** Possible transcriptional scenarios in response to binding by Myc and/or Mlx Network members. On the left, a hypothetical gene bearing an E box that is either of high- or low-affinity for Myc (++++ and +, respectively) (small gray box). It also contains a ChoRE of similar affinities for MondoA/ ChREBP-Mlx (large gray box). The gene is expressed at a low basal level in the absence of Myc and MondoA/ChREBP as might occur in quiescent cells maintained in low glucose. (**A**). In a low glucose environment, Myc may be induced in response to a normal proliferative signal. Myc-Max heterodimers bind to the high-affinity E box but not to the ChoRE and activate transcription in a glucose-independent manner. (**B**). In quiescent cells in a high glucose environment, MondoA/ChREBP-Mlx heterodimers enter the nucleus, bind to the high-affinity ChoRE but not to the E-box and activate transcription in a glucose-dependent manner. (**C**). In tumor cells that express excessively high Myc levels, Myc binds to both the high affinity E box and the low-affinity ChoRE. In the latter case, Myc-Max levels may be sufficiently high so as to exclude MondoA/ChREBP-Mlx binding entirely despite their otherwise high affinities for this site and particularly if they are expressed at low levels. Gene expression is now again glucose-independent. (**D**). In quiescent cells in which MondoA and/or ChREBP are particularly high, they may bind to both high-affinity ChoREs and low-affinity E boxes and regulate gene expression in a manner that is now glucose-dependent and Myc-independent. (**E**). In response to a normal proliferative stimulus and high-glucose, the gene is induced to a high level as a result of Myc-Max and MondoA/ChREBP-Mlx heterodimers each binding their respective high-affinity sites.

**Figure 3 cells-11-03974-f003:**
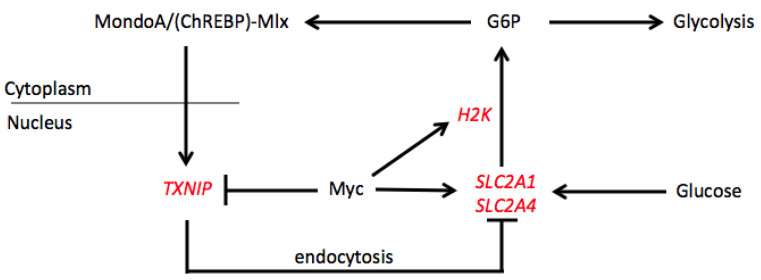
**The Myc and Mlx Networks cross-talk to modulate TXNIP and glycolysis**. The G6P-mediated cytoplasmic → nuclear transport of MondoA-Mlx heterodimers (and probably ChREBP-Mlx heterodimers as well) facilitates their binding to tandem ChoREs in the *TXNIP* gene’s promoter and up-regulates its expression [30,32,85,88]. Among the functions of TXNIP is to enhance endocytosis of the Glut1 and Glut4 glucose receptors encoded by *SLCA1* and *SLC2A4* genes [86,89]. Myc down-regulates *TXNIP* in part by displacing MondoA/ChREBP-Mlx heterodimers at ChoREs [87]. Myc also directly facilitates glucose uptake via binding to E boxes in the proximal promoters of the *SLC2A1* and *SLC2A4* genes and further drives glycolysis by a similar induction of the *H2K* gene, whose encoded protein, hexokinase 2, is one of the rate-limiting enzymes of glycolysis [52,65]. Genes that are direct targets for members of the Myc and/or Mlx Networks are shown in red.

**Figure 4 cells-11-03974-f004:**
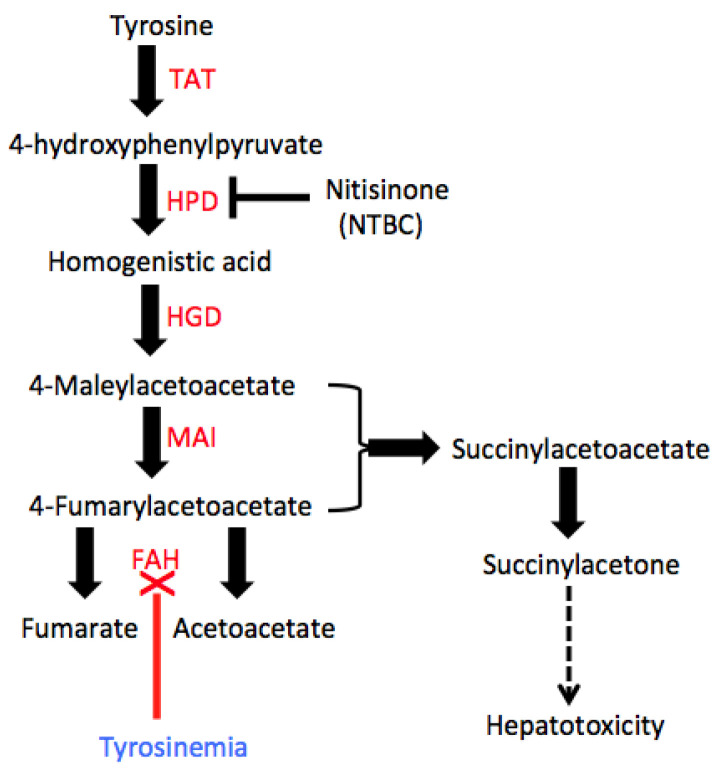
**Use of the FAH model of HT to assess long-term hepatocyte proliferation.** HT is caused by mutations in the *FAH* gene, which leads to accumulation of toxic tyrosine products, hepatocyte death and eventual liver failure [107]. Blocking the upstream enzyme HPD with Nitisinone (NTBC) prevents the formation of these catabolites and preserves hepatocyte viability and function. The disease process can also be reversed by transplanting donor *Fah+/+* hepatocytes via intrasplenic administration. Over 3–4 months, during which time Nitisinone is periodically discontinued and then resumed, recipient hepatocytes succumb to their accumulated toxic tyrosine by-products and are replaced by migrant donor hepatocytes. After a 50–100-fold expansion, donor cells typically replace as much as 70–80% of the recipient’s liver and allow for Nitisinone-free survival. The model also permits the delivery of two or more populations of competing donor hepatocytes whose ability to repopulate the recipient liver can be directly compared within the same animal. Abbreviations: TAT: tyrosine aminotransferase; HPD: 4-Hydroxyphenylpyruvate dioxygenase; HGD: Homogentisate 1,2-Dioxygenase; MAI: 4-Maleylacetoacetate isomerse; FAH: Fumarylacetoacetate hydrolase.

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
