# Peer review of "Regulation of Normal and Neoplastic Proliferation and Metabolism by the Extended Myc Network"

_cells, 2022, doi:10.3390/cells11243974_

Round 1

Reviewer 1 Report

This is an interesting review on “Regulation of normal and neoplastic proliferation and metabolism by the extended Myc network.  The author explored new insights on target gene co-regulation by two Networks: the Myc Network and the Mlx Network. The author shows the substantial regulatory overlap existing between this so-called “extended Myc network” and how it changes in ways that reflect the cells or tissue in which they are being studied, the conditions to which they are subjected and the stage of development.

The manuscript is generally well-written and structured.

Author Response

See attached pdf

Reviewer 2 Report

The review paper explores Myc and Mlx network, the overall interconnection between the two and the interplay in several process. The author presents a valid connection regarding the overlaps provided by recent papers, and links that could be explored in future studies and potential therapies.

General concept comments:

As stated by the author, the review is focused on Myc and Mlx network and their interplay in normal and neoplastic tissue. The strength of the concept was presented in author’s reference number 37, focused on hepatocytes. Also, 5 more references were later presented regarding Mlx. Although the author emphasized the aspect of Myc and Mlx on hepatocytes, therefore using it as reference for normal tissue, it would be interesting to know a bit more about Mlx in the immune response as several metabolic processes are also present in cell proliferation. Or at least a brief mention about studies involving Mlx in the metabolism of immune response if any.

            The comprehension about the interplay of the two networks could shed lights in future strategies specially involving cancer, for example. As the author stated, more studies are needed to clarify basic concepts of the biology of Myc and Mlx networks regarding its interplay and regulation. The literature regarding epigenetics and Myc regulation, specially involving microRNAs, has increased through the years. In that regard, it would be interesting if the author could briefly include the involvement of epigenetics in regulation of Myc and Mlx interplay, if there are evidence of microRNAs overlap that could contribute to fine-tune Myc and Mlx network interaction.

            In overall, the author presented a relevant topic that has been built up by his group and others in the field regarding the interplay of the two networks. As a continuous process, more studies are needed to refine the concept and to expand its relevance beyond fundamental biology, providing therapeutic insights.

            Although hit is not the focus of the criticism, the use of alphabetic letters to refer to paragraphs may create confusion. Instead, some topics could become normal paragraphs to introduce the following topic, making the text more fluidic.

Author Response

see attached pdf

Reviewer 3 Report

The present review titled “Regulation of Normal and Neoplastic Proliferation and Metabolism by The Extended Myc Network” is focused on implications of Myc network in proliferation control and metabolism of normal and neoplastic models. The Author, moreover, explores the correlation between Myc and Mlx Networks.

Overall, the manuscript is well written, even if some minor stylistic revisions should be made.

I think this review is acceptable for publication in Cells.

Author Response

See attached pdf

Reviewer 4 Report

This review well summarized the regulation of Myc and Mlx networks in normal and neoplastic cells. To my impression, it can be accepted in its present form. I also have some comments for reference.

1) I suggest the author lists all experimentally validated coregulated genes by Myc and Mlx networks in a table, which will be a good guidance for the future network regulation in cancer gene therapy.

2) It will be very nice to discuss the potential of Myc and Mlx networks-based therapy in cancer therapy.

3) The relationship between Myc/Mlx networks with tumor suppressors like TP53 was necessary to be described in more details. The co-regulating both TP53 and Myc/Mlx Networks might be an efficient anti-cancer strategy.

4) Based on the regulation of Myc and Mlx networks in normal and neoplastic cells, the cancer therapy based on this regulation could be further discussed.

Author Response

see attached pdf
